# Predictive Modeling for Suicide-Related Outcomes and Risk Factors among Patients with Pain Conditions: A Systematic Review

**DOI:** 10.3390/jcm11164813

**Published:** 2022-08-17

**Authors:** Shu Huang, Motomori O. Lewis, Yuhua Bao, Prakash Adekkanattu, Lauren E. Adkins, Samprit Banerjee, Jiang Bian, Walid F. Gellad, Amie J. Goodin, Yuan Luo, Jill A. Fairless, Theresa L. Walunas, Debbie L. Wilson, Yonghui Wu, Pengfei Yin, David W. Oslin, Jyotishman Pathak, Wei-Hsuan Lo-Ciganic

**Affiliations:** 1Department of Pharmaceutical Outcomes and Policy, College of Pharmacy, University of Florida, Gainesville, FL 32610, USA; 2Department of Population Health Sciences, Weill Cornell Medicine, New York, NY 10065, USA; 3Health Science Center Libraries, University of Florida, Gainesville, FL 32610, USA; 4Department of Health Outcomes and Biomedical Informatics, College of Medicine, University of Florida, Gainesville, FL 32610, USA; 5Cancer Informatics Shared Resource, University of Florida Health Cancer Center, University of Florida, Gainesville, FL 32610, USA; 6Division of General Internal Medicine, School of Medicine, University of Pittsburgh, Pittsburgh, PA 15213, USA; 7Center for Health Equity Research Promotion, Veterans Affairs Pittsburgh Healthcare System, Veterans Health Administration, Pittsburgh, PA 15240, USA; 8Center for Drug Evaluation and Safety, College of Pharmacy, University of Florida, Gainesville, FL 32610, USA; 9Division of Health and Biomedical Informatics, Department of Preventive Medicine, Feinberg School of Medicine, Northwestern University, Chicago, IL 60611, USA; 10Department of Psychiatry, University of Florida, Gainesville, FL 32610, USA; 11Department of General Internal Medicine and Geriatrics, Feinberg School of Medicine, Northwestern University, Chicago, IL 60611, USA; 12Veterans Integrated Service Network 4 Mental Illness Research, Education, and Clinical Center (MIRECC), Corporal Michael J. Crescenz VA Medical Center, Philadelphia, PA 15240, USA; 13Perelman School of Medicine, University of Pennsylvania, Philadelphia, PA 19104, USA

**Keywords:** suicide-related outcomes, pain conditions, predictive modeling

## Abstract

Suicide is a leading cause of death in the US. Patients with pain conditions have higher suicidal risks. In a systematic review searching observational studies from multiple sources (e.g., MEDLINE) from 1 January 2000–12 September 2020, we evaluated existing suicide prediction models’ (SPMs) performance and identified risk factors and their derived data sources among patients with pain conditions. The suicide-related outcomes included suicidal ideation, suicide attempts, suicide deaths, and suicide behaviors. Among the 87 studies included (with 8 SPM studies), 107 suicide risk factors (grouped into 27 categories) were identified. The most frequently occurring risk factor category was depression and their severity (33%). Approximately 20% of the risk factor categories would require identification from data sources beyond structured data (e.g., clinical notes). For 8 SPM studies (only 2 performing validation), the reported prediction metrics/performance varied: C-statistics (*n* = 3 studies) ranged 0.67–0.84, overall accuracy(*n* = 5): 0.78–0.96, sensitivity(*n* = 2): 0.65–0.91, and positive predictive values(*n* = 3): 0.01–0.43. Using the modified Quality in Prognosis Studies tool to assess the risk of biases, four SPM studies had moderate-to-high risk of biases. This systematic review identified a comprehensive list of risk factors that may improve predicting suicidal risks for patients with pain conditions. Future studies need to examine reasons for performance variations and SPM’s clinical utility.

## 1. Introduction

Suicide was the 10th leading cause of death in the United States (USA) in 2019 and remains a major public health concern [1]. The age-adjusted suicide rate in the US increased by 35% from 10.5 per 100,000 population in 1999 to 14.2 in 2018 [2,3]. Patients with pain conditions have elevated risks. For example, the prevalence of suicidal ideation among individuals with chronic pain is approximately three times as high as in those without [4,5]. Furthermore, increasing rates of suicide have coincided with the epidemic of nonmedical opioid use and overdose over the past two decades [6]. Together, suicide and opioid overdose contribute to excess adverse health consequences [7], with growing evidence indicating opioid use disorder (OUD), opioid overdose, and suicide are associated [6].

The relationship between pain conditions/severity, pain management (including prescription opioids), OUD, opioid overdose, and suicide is complex [8,9,10,11]. Individuals with OUD, opioid overdose, and suicide share common risk factors, such as mental health disorders. OUD is a predominant precursor to overdose and suicide deaths [12,13,14]. Other potential interactions of opioid use and suicide among patients with pain conditions include patients suffering from pain being more likely to develop suicidal ideations from suppression and sensitivity of their neural reward pathways [10,15].

Given the unique vulnerabilities of patients with pain conditions and complex interactions of predisposing and enabling factors in this population, suicide prediction models (SPMs) may be effective tools to identify patients at risk and prevent adverse outcomes, including death [16,17]. Although previous SPM studies have shown promising results, few have focused on patients with pain conditions. Belsher et al.’s systematic review evaluated predictive accuracy of SPM studies from 2015 to 2018, focusing on the general population [18]. The existing SPMs had good C-statistics for suicide deaths, but very low positive predictive value (PPV), and the overall predictive performance remained poor for the general population. To develop and improve the prediction performance for suicide-related outcomes in patients with pain conditions, this study aimed to conduct a systematic review to (1) to evaluate the performance of existing SPMs among patients with pain conditions, and (2) to identify risk factors and derived data sources of suicidal ideation (SI), suicide attempts (SA), suicide deaths (SD), and suicide behaviors (SB) [19,20] among patients with pain conditions [21,22].

## 2. Materials and Methods

### 2.1. Data Sources and Search Strategies

This systematic review followed internationally accepted gold standard guidelines stated in the Cochrane Handbook for Systematic Reviews of Interventions and was complied with Preferred Reporting Items for Systematic Reviews and Meta-analyses (PRISMA) guidelines [23,24]. The study protocol was registered with PROSPERO (i.e., International Prospective Register of Systematic Reviews; Registration ID: CRD42020215887). An initial search was performed on 13 January 2021 of grey literature sources and the bibliographic databases PubMed, PsycINFO, EMBASE, SCOPUS, Cochrane Library, Web of Science, ProQuest Thesis Dissertations, and CINAHL for studies published from January 2000 to December 2020. According to the Cochrane Handbook, to reduce the risk of publication bias and retrieve all relevant evidence, the inclusion of gray literature sources such as conference abstracts and proceedings, theses, dissertations, and reports are necessary [23]. Additionally, we manually screened the reference lists from published suicide-related systematic reviews.

In consultation with a librarian (L.E.A.), the search strategy combined database-specific controlled vocabulary truncated and phrase-searched keywords in titles and abstracts as available for the concepts pain conditions, suicide-related outcomes, and risk prediction or predictive models. Search results were limited to the English language. Appendix A provides a complete list of search queries for databases used.

### 2.2. Study Selection and Data Extraction

We restricted our search to observational studies identifying risk or protective factors associated with suicide-related outcomes for adult patients aged ≥18 years with any pain condition. Since most studies did not specify pain conditions or differentiate between acute and chronic pain, we included studies focusing on patients with physical pain (e.g., fibromyalgia) and/or psychological pain. Psychological pain was defined as having intense emotional pain, often associated with psychiatric disorders (e.g., depression) or emotional trauma [25,26]. We excluded case reports, opinions, animal research, and commentaries, as well as interventional studies intended to evaluate interventional efficacy (e.g., ketamine use) on suicide-related outcomes [27,28]. Eligible studies were determined to be SPM studies if they described all of the following: (1) data sources and predictor candidates (or features), (2) modeling methods or prediction procedures (e.g., randomly split data into training and testing), and (3) SPM’s prediction performance (e.g., discrimination and calibration measures) [29]. Discrimination measures the extent to which predicted high-risk patients exhibit higher rates of suicide-related outcomes compared to those predicted as low risk (e.g., C-statistic) [30]. Calibration measures risk estimates’ accuracy, considering agreement between suicide-related events estimated and observed numbers [31]. Given our goal was identifying SPM studies or studies reporting suicide risk factors for patients with pain conditions, we excluded purely descriptive studies without any regression analysis or otherwise adjusted analysis.

After a comprehensive literature search and removal of duplicates, three investigators (S.H., M.O.L., P.Y.) double screened the articles’ titles and abstracts, and independently screened for inclusion and exclusion eligibility based on full text using Covidence (Melbourne, Australia) [32]. We extracted study information and details using a standardized Microsoft Excel (Redmond, WA, USA) form. We extracted information regarding author, year of publication, country, type of study, data source, patient population (e.g., sample size, age, pain conditions), outcomes of interest and definitions, outcome rates, statistical or prediction modeling methods, and significant risk or protective factors and risk estimates (e.g., point estimate of risk ratios [RR] or odds ratios [OR], 95% confidence intervals [95%CI], and *p*-values). We categorized data-source types identifying and measuring each risk factor into: (1) “structured data” that naturally occur (e.g., clinical documentation or billing activities results) and outside a research context such as prespecified fields using standardized terminologies or codes in electronic medical records (EMR) or administrative claims data; (2) “unstructured data” included text-based clinical notes and reports in EMR requiring detailed manual review or sophisticated machine learning technologies such as natural language processing (NLP) to extract information; or (3) “collected data” that are not found in the EMR and require linkage to additional data sources such as a survey or registry. For studies developing an SPM, we collected prediction performances, including C-statistic, accuracy, sensitivity, and PPV. Disagreements between reviewers were resolved by consultation with a fourth investigator (W.-H.L.-C.).

### 2.3. Risk of Bias Assessment

Using a modified Quality In Prognosis Studies (QUIPS) tool, two investigators (S.H., M.O.L.) independently assessed included SPM studies’ quality using bias domains including (1) study participation, (2) study attrition, (3) prognostic factor measurement, (4) outcome measurement, (5) study confounding, and (6) statistical analysis and reporting [33]. We chose QUIPS, because its bias assessments were designed for prognostic modeling studies and were more appropriate for SPM study designs. We rated each SPM study in the 6 potential bias domains as having low, moderate, or high risk of bias. A third investigator (W.L-C.) led discussions to resolve disagreements.

### 2.4. Study Outcomes of Interest

Our outcome of interest was any suicide-related outcome including SI, SA, SD, and any other SB. SI (or suicidal thoughts) is thinking or planning to commit suicide, but without taking any action [34]. SA is when someone harms themself with the intent to end their life, which may or may not result in death [35]. SD is mortality resulting from self-injury with the intent to end one’s life [35]. SB includes SI, SA, and SD [35]. For included SPM studies, given suicide-related events’ rarity, C-statistics will not account for outcome prevalence information and can overestimate an SPM’s utility [36]. Thus, we extracted other prediction performance metrics reported such as overall accuracy or misclassification rate, sensitivity, or recall (measured as proportion of patients having suicide-related outcomes correctly identified as being at risk), PPV or precision (measured as proportion of patients identified as being at risk of suicide who have suicide-related outcomes), and negative predictive values (NPV) (i.e., fraction of patients identified with no or low suicide risk lacking suicide-related outcomes) [37].

## 3. Results

### 3.1. Characteristics of Included Studies

Figure 1 shows the initial literature search yielded 1802 records, of which 1579 remained after removing duplicates. Title and abstract screening resulted in excluding articles unrelated to the study objective (*n* = 1294) or duplicates (*n* = 3). We then identified four additional studies from prior suicide-related systematic reviews’ references, which led to 286 full-text studies assessed for eligibility. Full-text review further excluded 199 studies for the following: (1) not target population/no pain conditions (*n* = 109), (2) descriptive studies without any regression analysis or otherwise adjusted analysis (*n* = 31), (3) commentary, abstract, or no full-text (*n* = 21), (4) ineligible study design (*n* = 21), (5) no suicide-related outcome (*n* = 12), and (6) no predictor/risk factor identified (*n* = 5). This systematic review identified 87 studies reporting suicide risk factors among patients with pain conditions, including eight studies developing SPMs.

Table 1 and Appendix A summarize characteristics for each study. Sixty-four percent (*n* = 56) of the 87 included studies were cross-sectional, followed by retrospective cohort studies (*n* = 14, 16%), and case-control studies (*n* = 12, 14%). Sample sizes ranged from 50 to 4,863,086. Nearly half were conducted in the USA (*n* = 38, 44%), followed by South Korea (*n* = 7, 8%), Canada (*n* = 6, 7%), and the United Kingdom (UK) (*n* = 6, 7%). Eleven studies (13%) included cancer patients [38,39,40,41,42,43,44,45,46,47,48]. The most common physical pain conditions specified included fibromyalgia, cancer pain, and back pain/low back pain. Besides patients with pain conditions (physical pain and/or psychological pain), many included studies featured special populations such as veterans and older adults. The most common data sources used were multisite questionnaire (*n* = 28, 32%), single-site questionnaire (*n* = 25, 29%), and EMR data (*n* = 19, 22%).

### 3.2. Factors Associated with Suicide-Related Outcomes among Patients with Pain Conditions

Table 2 and Appendix A summarize identified suicide-related outcome risk factors among patients with pain conditions. We further grouped risk factors into 27 categories based on characteristics and clinical conditions (Appendix A). The top ten most frequently reported risk factor categories were: depression/depressive disorders and their severity (reported by 33% of included studies), other patient reported factors (e.g., sexual/physical abuse, hopelessness; 29%), any unspecified physical health illness or comorbidity index (24%), other mental health conditions (15%), pain duration/severity/intensity (15%), anxiety disorders and their severity (14%), other specific pain conditions (13%), history of suicide-related outcomes (9%), sleep disorders including insomnia (9%), and social determinants of health (9%). Only six (7%) studies identified opioid use/dosage as a risk factor for suicide among patients with pain conditions. Based on our analysis, 67% of all risk factors were routinely identifiable in structured data, 19% were available primarily from unstructured data, and the remaining 15% of risk factors were potentially identifiable from both.

Depression was the top reported risk factor, and was also considered a type of psychological pain when selecting eligible studies. To avoid tautology, we checked the four articles that included patients with depression [49,50,51,52]. Three studies included patients with depression only, and identified risk factors other than depression [49,50,51]. One study included depressed patients and healthy controls, and therefore identified depression as a suicide risk factor [52]. Among the remaining 83 studies, including patients without depression as study cohort, 28 identified depression as a risk factor focused on patients with physical/psychological pain conditions other than depression. Therefore, depression was considered as an independent suicide risk factor.

### 3.3. Performance of Studies Developing Suicide Prediction Models

Table 3 summarizes the eight SPM studies’ main characteristics [19,20,53,54,55,56,57,58]. None comprehensively reported model prediction performance metrics. Three reported C-statistics, which ranged from 0.67 to 0.84 [19,20,57]. Five reported overall accuracy, which ranged from 0.78 to 0.96 [53,54,55,56,58]. Two reported sensitivities, which ranged from 0.65 to 0.91 [19,20]. Three reported PPV, which ranged from 0.01 to 0.43 [19,20,57]. Two (25%) validated their predictive model using external validation and bootstrap cross-validation methods [57,58]. Two used structured EMR to develop SPMs (C-statistics: 0.67–0.82; PPV: 0.01–0.14) [19,57]. Table 3 presents QUIPS risk of bias assessment results. The quality of the eight SPM studies for pain patients varied. For four studies (50%) [19,56,57,58], all risk of bias domains were rated low. The remaining four had at least one risk of bias domain rated moderate or high risk [20,53,54,55].

One research team (Fishbain et al.) conducted half of the studies [53,54,55,56], three of which used the same study sample (~2000 USA community and rehabilitation facility residents) [53,54,55]. Using a 600-item survey, they developed multivariable logistic regression models to predict various suicide-related outcomes including SI and SB with an overall accuracy ranging from 0.78 to 0.96. Several QUIPS bias domains in the three studies were rated moderate or high risk due to the potential for selection bias and attrition bias in the study sample, lack of definitions for suicide-related outcome measures, inadequate adjustment for potential confounders, and insufficient statistical reporting. The fourth SPM study (Fishbain et al. 2009) [56] developed a multivariate logistic regression model to predict SI risk among smokers with chronic low back pain (*n* = 81) from a single pain facility, with an overall accuracy of 78%. Risk of bias across all domains for this study was low. However, none of these four performed a validation analysis of their SPMs.

Lopez-Morinigo et al. [19] conducted a retrospective cohort study using single-site EMR data to predict suicide deaths among UK patients receiving secondary mental health care (referred mental health care) (*n* = 13,758). SDs were identified from death certificates using International Classification of Diseases, Tenth Revision, Clinical Modification (ICD-10-CM) diagnosis codes. Using multivariable Cox proportional hazards regression models, the SPM C-statistic was 0.67, with 0.65 sensitivity and 0.01 PPV at the balanced risk threshold. This study had low risk of bias but did not perform SPM validation.

McKernan et al. [57] conducted a case-control study predicting SI and SA in 8879 patients with fibromyalgia using EMR data. Using bootstrapped L-1 penalized regression, C-statistics were 0.80 for SI and 0.82 for SA. However, PPV remained low for SI (0.14) and SA (0.08). Risk of bias across all domains for this study was low. This study used an independent sample to evaluate published SPMs’ external validity.

Sun et al.’s [20] cross-sectional study used chart review and survey data to develop hierarchical regression and multivariable logistic regression models to predict post suicidal ideation and suicide attempts among 137 patients with major depressive disorders in China. At the balanced cutoff threshold, C-statistic was 0.84. This study had high risk of bias for the prognostic factor measurement domain due to unclear temporality of the suicide-related outcome’s measurement, and low risk biases for remaining domains. It did not perform SPM validation.

Tektonidou et al. [58] conducted a cross-sectional study using 2007–2008 National Health and Nutrition Examination Survey to identify US adults aged ≥40 years with arthritis, diabetes, or cancer (*n* = 2344). They developed a random forest model with bootstrapping and cross-validation methods to predict suicidal ideation among this high-risk group. The prediction metric reported in the study was 0 misclassification error (i.e., 100% accuracy). This study had low risk of bias.

## 4. Discussion

In this systematic review, including 87 studies reporting suicide risk factors, we identified 107 risk factors associated with suicide-related outcomes among patients with pain conditions. Two-thirds of these risk factors are identifiable from data routinely collected for clinical documentation or billing purposes, such as structured data within an EMR. Only eight studies developed SPMs to predict suicide-related outcomes for patients with pain conditions, of which two used structured data sources. Given suicide-related outcomes’ rarity, PPVs were low, as expected. Most SPM studies had major limitations including lacking rigorous study designs, lacking robust processes for SPM development (e.g., validation), lacking thorough reporting of prediction metrics, and risk of biases. One SPM study by McKernan et al. [57] had low risk of biases and provided good quality of evidence.

Risk of SD was at least doubled in patients with chronic pain compared to those without [59]. Consistent with prior studies predicting suicide-related outcomes in general populations, common suicide risk factors for patients with pain conditions identified in our review included depressive disorders, unspecified physical or somatic pain conditions, and anxiety disorders. Unique risk factors for patients with pain conditions identified included migraine/headaches, fibromyalgia, opioid use and dosage, and perceived burdensomeness. Of the identified risk factors, modifiable risk factors valuable for designing targeted interventions are physical health conditions and mental health disorders, which may be managed with medication, behavioral therapy, and other treatment. Furthermore, this study provided a comprehensive list of risk factors critical for improving future SPM studies for identify individuals at risk of suicide among patients with pain conditions [60].

Nearly 20% of risk factors (e.g., perceived hopeless) identified in our review are only collected from survey or unstructured EMR data (e.g., clinical notes) [61]. Developing and administering a validated questionnaire instrument for patients with pain is time-consuming. Unstructured EMR data often contain valuable information for prediction such as patient behaviors and social and behavioral determinants of health infrequently captured in structured data. With advancements in computational linguistics and machine learning, NLP systems are capable of processing free-text clinical notes and producing structured outputs useful as predictors [62]. Identify additional features from unstructured clinical notes using NLP may improve SPMs’ prediction accuracy [63].

No SPM study identified in our review thoroughly reported prediction metrics unbiasedly [29,64]. Relying on C-statistics to predict rare outcomes can be misleading and uninformative [64]. Consistent with Belsher et al.’s [18] systematic review of SPMs in the general population, rarity of suicide-related outcomes in patients with pain conditions led to low PPVs (5/11 studies reported models with PPVs > 0.01) in the included SPM studies. Despite very high sensitivity and specificity, low PPVs limit a model’s clinical utility because of false positives [65]. Applying advanced machine learning methods such as neural networks may address shortcomings of current SPMs and improve identification of patients with pain conditions at high-risk of suicide-related outcomes for targeted interventions. This will require large sample sizes, which may only be attainable with large EMR- or claims-based datasets linked to death records.

Risk of bias assessments identified several major weaknesses in existing SPMs for patients with pain conditions. In total, 75% of the SPM studies used cross-sectional designs, which lack temporality between exposures or predictors and outcomes [66]. For example, Sun’s study did not clearly specify measuring risk factors before suicide-related outcomes [20]. When developing SPMs, longitudinal and prognostic modeling designs better simulate continuous population screening [29]. Furthermore, most SPM studies did not conduct any validation analyses to avoid model overfitting. For the two SPM studies that conducted validation, the performance of the model was not clearly reported. For example, Tektonidou et al. [58] developed random forest models, with a misclassification error of 0 reported for cross-validated test sets. Achieving perfect prediction in real-world settings is implausible, especially for rare suicide-related outcomes. Future studies should include robust validation processes to ensure developed SPMs are well-calibrated and reproducible in real-world data. Future SPM studies’ should make their data and methods publicly available according to Findability, Accessibility, Interoperability, and Reuse (FAIR) principles of digital assets.

Our study had two major limitations. First, we excluded studies and theses not written in English. Second, because of the limited number of SPM studies identified for patients with pain conditions, mixed study designs, and effect estimates’ varied reporting, we could not conduct pooled analyses combining the eight SPM studies’ results.

We propose the following recommendations for future studies to develop SPMs for patients with pain conditions. First, using a composite measure of any suicide-related outcomes in SPM may increase event frequency and improve PPVs, but might limit the utility of predicting suicide deaths. Second, an SPM’s utility is influenced by availability of important predictors that may require better data collection in routine clinical care or the billing process. Third, thoroughly reporting different prediction metrics to allow researchers to evaluate SPMs in an unbiased manner is important [67]. SPMs’ development should include validation processes to minimize overfitting and increase model calibration ability, and to evaluate performance variations across subpopulations and multiple datasets. Fourth, although opioid use, OUD, overdose, and suicide may impact or interact among patients with pain conditions, only a few studies in our review included opioid use as a suicide risk factor, and no studies have examined OUD or overdose history as risk factors. An aggressive prescription opioid supply regulation to curb the opioid crisis can result in unintended consequences of patients with pain no longer having access to adequate pain management (including prescription opioids) and thus lead to an increased risk for depression and suicide. Future studies should include opioid use and OUD and overdose history when developing SPMs. Finally, clinical and policy utility of SPMs is influenced by their outcome risk and benefit profile, interventions type, and resource availability. Using SPMs with proper risk stratification (e.g., top 1st percentile, top 5th percentile as high risk) may more effectively identify high-risk individuals to better target time-sensitive interventions. For rare outcomes and SPMs with low PPVs, additional screening and assessment are needed to avoid unintended consequences resulting from false positives.

## 5. Conclusions

Although our findings revealed major limitations of existing SPMs and the need for mature and robust predictive models to predict suicide-related outcomes for patients with pain conditions, the systematic review provides a comprehensive summary of evidence of the risk factors of suicide-related outcomes and SPMs for patients with pain conditions that can be used to improve SPMs’ performance. Risk factors identified from this review may be valuable for clinicians to identify patients with elevated suicide risks for target interventions.

## Figures and Tables

**Figure 1 jcm-11-04813-f001:**
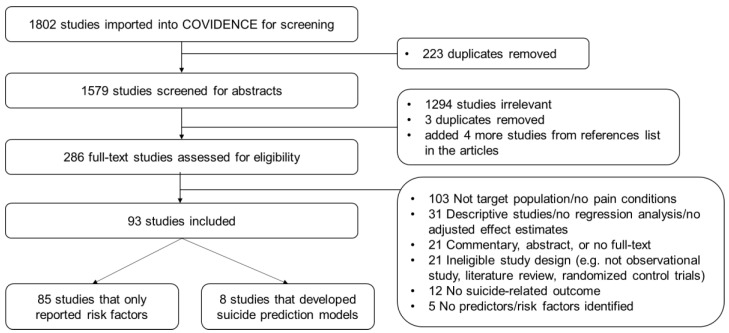
Preferred Reporting Items for Systematic Reviews and Meta-Analyses (PRISMA) Flowchart of the Systematic Review.

**Table 1 jcm-11-04813-t001:** Characteristics of Suicide Prediction Modeling Studies.

Author, Year	Country	Study Design	Type of Data Sources	Study Population ^a^	Total # Pts	Outcome (s)	Statistical Methods	Validation	C-Statistic	Accuracy	Sensitivity	PPV
Fishbain, 2009	USA	Cross-sectional	Single site questionnaire	Chronic low back pain pts who smoke	81	SI	Logistic regression	No validation	N/A	0.78	N/A	N/A
Fishbain, 2011	USA	Cross-sectional	Community questionnaire (multisite)	Rehabilitation pain pts	2264	SI	Logistic regression	No validation	N/A	0.96	N/A	N/A
Fishbain, 2012	USA	Cross-sectional	Community questionnaire (multisite)	Rehabilitation pain pts	2264	SI ^b^	Logistic regression	No validation	N/A	0.78–0.88	N/A	N/A
Fishbain, 2012	USA	Cross-sectional	Community questionnaire (multisite)	Rehabilitation pain pts	2264	SB	Logistic regression	No validation	N/A	0.87–0.95	N/A	N/A
Lopez-Morinigo, 2018	UK	Retrospective cohort	Single site EMR	Pts seen in a comprehensive pain clinic	13,758	SD	Cox proportional hazards model	No validation	0.67	N/A	0.65	0.01
McKernan, 2018	USA	Case-control	Single site EMR	Pts with fibromyalgia	8879	SI & SA	Bootstrapped L-1 penalized regression	Independent sample to test the external validation of published SPMs	0.82 (SA), 0.80 (SI)	N/A	N/A	0.08 (SA), 0.14 (SI)
Sun, 2020	China	Cross-sectional	Single site chart review, Single site questionnaire	Psychiatric outpatients with major depressive disorder	137	Past SI & SA	Logistic regression	No validation	0.84	N/A	0.91	0.43
Tektonidou, 2011	USA	Cross-sectional	Nationwide questionnaire	Pts aged ≥40 with arthritis, diabetes, or cancer	2344	SI	Random forest model	Bootstrap, Cross-validation	N/A	1 ^c^	N/A	N/A

Abbreviations: EMR: electronic medical record, N/A: not available, pts: patients, PPV: positive predictive value, SA: suicidal attempts, SD: suicide deaths, SI: suicidal ideation, UK: United Kingdom, USA: United States of America. ^a^ All patients are adult patients. ^b^ SI related item: prefer death over disability ^c^ The cross-validated test set misclassification error for each random forest was 0.

**Table 2 jcm-11-04813-t002:** Summary of Individual Risk Factors Identified from more than 3 studies by Data Source for Identification ^a^.

Risk Factors	Number of Studies	% of the 87 Studies	Data Source that can be Used to Identify Risk Factors ^b^
Depression/depressive disorders and their severity	29	33%	Structured/Unstructured/Collected data ^c^
Any unspecified physical or somatic pain conditions	17	19%	Structured
Anxiety disorders and their severity	12	14%	Structured/Unstructured/Collected data
History of suicidal behavior/ideation/attempts/suicidality	8	9%	Structured/Unstructured/Collected data
Pain duration/severity/intensity	8	9%	Unstructured/Collected data
Sleep disorders including insomnia	8	9%	Structured
Age	7	8%	Structured
Psychache/mental pain	7	8%	Unstructured/Collected data
PTSD	6	7%	Structured
Fibromyalgia pain	5	6%	Structured
Gender	5	6%	Structured
Migraine/headaches and frequency	5	6%	Structured
Opioid use and dosage (e.g., >100 MME)	5	6%	Structured
Perceived burdensomeness	5	5%	Unstructured/Collected data
Antidepressant use and type	4	5%	Structured
Comorbidity or comorbidity index	4	5%	Structured
Perceived/feeling hopeless	4	5%	Unstructured/Collected data
Race/ethnicity	4	5%	Structured
AUD	3	3%	Structured
Anger issues	3	3%	Structured/Unstructured/Collected data
Any mental health illness	3	3%	Structured
Any unspecified physical health illness	3	3%	Structured
Back pain/low back pain	3	3%	Structured
Cancer pain	3	3%	Structured/Unstructured/Collected data
Drug use disorders	3	3%	Structured
History of sexual/physical abuse	3	3%	Structured/Unstructured/Collected data
Marital status (e.g., unmarried)	3	3%	Structured/Unstructured/Collected data
Mental quality of life	3	3%	Unstructured/Collected data
Pain catastrophizing	3	3%	Unstructured/Collected data
Perceived/feeling stressful	3	3%	Unstructured/Collected data
Respiratory diseases	3	3%	Structured
Unemployment	3	3%	Unstructured/Collected data

Abbreviations: AUD: alcohol use disorder, EMR: electronic medical records, MME: morphine milligram equivalents, PTSD: Posttraumatic stress disorder. ^a^ Risk factors reported from less than 3 studies are listed in Appendix A. ^b^ This is the authors’ view of where the majority of these data can be captured from. ^c^ We categorized the type of data sources that can be used to identify and measure each risk factor into: (1) “structured data” that naturally occur (e.g., as a result of clinical documentation or billing activities) and outside a research context such as structured EMR or administrative claims data; (2) “unstructured data” include unstructured clinical notes in EMR required efforts such as natural language processing to extract information; and (3) “collected data” that require additional design such as from a questionnaire or registry. Structured/Unstructured/Collected data refers to some of the risk factors (e.g., depression diagnosis) and may be identified from structured data, and some (e.g., depression severity) may be identified from unstructured data or questionnaires.

**Table 3 jcm-11-04813-t003:** Quality In Prognosis Studies (QUIPS) Risk of Bias Assessment Results.

	Study Participation	Study Attrition	Prognostic Factor Measurement	Outcome Measurement	Study Confounding	Statistical Analysis and Reporting
Fishbain, 2009	Moderate	Moderate	Low	High	Moderate	Moderate
Fishbain, 2011	High	Moderate	Low	Moderate	Moderate	Moderate
Fishbain, 2012	High	Moderate	Low	Moderate	Moderate	Moderate
Fishbain, 2012	Low	Low	Low	Low	Low	Low
Lopez-Morinigo, 2018	Low	Low	Low	Low	Low	Low
McKernan, 2018	Low	Low	Low	Low	Low	Low
Sun, 2020	Low	Low	Moderate	Low	Low	Low
Tektonidou, 2011	Low	Low	Low	Low	Low	Low

## Data Availability

The data are publicly available.

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
