# Peer review of "Predictive Modeling for Suicide-Related Outcomes and Risk Factors among Patients with Pain Conditions: A Systematic Review"

_jcm, 2022, doi:10.3390/jcm11164813_

Round 1
Reviewer 1 Report
I think this is really meaningful review study. However, there is still minor issues, so, I would be appreciated to revise it as followed. 1) I think it would be better not to mention the limitations of this review study at the end of the conclusion, so I think it would be better to revise it as follows."Although our findings revealed major limitation of existing SPMs and need for mature and robust predictive model to predict suicide -related outcomes for patients with pain conditions, the systematic review provides a comprehensive summary of evidence on the risk factors of suicide-related outcomes and SPMs for patients with pain conditions".
2) At the last part of the conclusion, could you please add how the findings obtained from this review study can be used in clinical practice in the future?Author Response
Please see the attachment.

Reviewer 2 Report
The methodology followed international guidelines. The field of investigation was extended to gray literature. The eligibility criteria are well defined and strict. From the large amount of data, the authors obtained 87 studies, which were organized according to risk factors. The authors identified 8 predictive studies. The article reports some corrections in the results, the result of a previous review. The discussion appears to be consistent with the results.
Author Response
Please see the attachment.

This manuscript is a resubmission of an earlier submission. The following is a list of the peer review reports and author responses from that submission.
Round 1
Reviewer 1 Report
This is a methologically sound study with careful attention to the complexity of the analyses as well as the variability of the sources. However it is not clear if the authors risk a tautology by selecting studies in which depession without other indicators of physical pain is included. This is of concern because depression was the largest variable in the SPM. The conclusion that existing pain related predictive models are inadequate is amply demonstrated.